# Origin and segregation of the human germline

Aracely Castillo-Venzor[1,2,3,4,*] , Christopher A Penfold[1,2,3,4,*] , Michael D Morgan[7,8,*] , Walfred WC Tang[1,3,4] ,
Toshihiro Kobayashi[5,6], Frederick CK Wong[1,3,4], Sophie Bergmann[2,3,4], Erin Slatery[2,3,4] , Thorsten E Boroviak[2,3,4],
John C Marioni[7,8,9], M Azim Surani[1,2,3,4]

**Human germline–soma segregation occurs during weeks 2–3 in gastrulating embryos. Although direct studies are hindered, here, we investigate the dynamics of human primordial germ cell (PGC) specification using in vitro models with temporally resolved single-cell transcriptomics and in-depth characterisation using in vivo datasets from human and nonhuman primates, including a 3D marmoset reference atlas. We elucidate the molecular signature for the transient gain of competence for germ cell fate during peri-implantation epiblast development. Furthermore, we show that both the PGCs and amnion arise from transcriptionally similar TFAP2A-positive progenitors at the posterior end of the embryo. Notably, genetic loss of function experiments shows that TFAP2A is crucial for initiating the PGC fate without detectably affecting the amnion and is subsequently replaced by TFAP2C as an essential component of the genetic network for PGC fate. Accordingly, amniotic cells continue to emerge from the progenitors in the posterior epiblast, but importantly, this is also a source of nascent PGCs.**

## Introduction

Human primordial germ cells (PGCs) are among the first lineages to emerge in the developing gastrulating peri-implantation embryo at weeks 2–3, eventually developing into sperm or eggs. The parental gametes generate the totipotent state at fertilisation and transmit genetic and epigenetic information necessary for development.

The specification of PGCs is linked with the initiation of the unique germ cell transcriptomic and epigenetic programs (Guo et al, 2015; Irie et al, 2015, 2018; Tang et al, 2015, 2016, 2022; Kobayashi et al, 2017). Aberrant specification and development of germ cells can lead to sterility, germ cell-derived cancers, and other human diseases with long-term consequences across generations. Ethical and technical reasons restrict direct studies on nascent human PGCs, necessitating in vitro models, which are, however, experimentally tractable for mechanistic insights (Irie et al, 2015, 2018; Sasaki et al, 2015; Tang et al, 2016, 2022; Kobayashi et al, 2017). Because of the in vitro nature of these models, comprehensive comparisons with rare human embryos and animal proxies, including in vivo development of nonhuman primates such as marmosets, can be significantly informative for germline biology.

The induction of PGC-competent cells from cultured pluripotent stem cells (PSCs) is possible using self-renewing cells cultured in a 4-inhibitor medium (4i conditions) (Irie et al, 2015) or transient pre-mesendoderm (iMELC/PreME) populations (Sasaki et al, 2015; Kobayashi et al, 2017). The timing and regulation of the transient state of competence for PGC fate in human embryos are not yet fully defined but likely determine the number of founder PGCs in vivo. If aggregated into 3D embryoid bodies, these competent cells give rise to 10–40% PGC-like cells (PGCLCs) in response to BMP and other cytokines (Irie et al, 2015; Sasaki et al, 2015; Kobayashi et al, 2017). The remaining cells adopt somatic fates, but their relationship with the emerging PGCLCs remains unclear (Irie et al, 2015; Sasaki et al, 2015; Kobayashi et al, 2017). Defining the characteristics of the somatic lineages in embryoid bodies may help identify soma–PGC interactions and reveal the context of how PGCs form in experimental models concerning the lineages in the embryo.

In vitro models identified SOX17, PRDM1, and TFAP2C as the core regulators of human PGC fate (Irie et al, 2015; Sasaki et al, 2015; Kobayashi et al, 2017; Kojima et al, 2017). This tripartite network for PGC fate has also been observed in vivo in other species that develop as a bilaminar disc, including cynomolgus, marmosets, rabbits, and pigs (Sasaki et al, 2016; Sybirna et al, 2019; Alberio et al, 2021; Kobayashi et al, 2021; Zhu et al, 2021; Bergmann et al, 2022).

On the other hand, Sox17 is not a critical regulator of PGC specification in rodents where the embryos develop as egg cylinders (Kanai-Azuma et al, 2002). Notably, when SOX17 is the critical

[1]Wellcome Trust/Cancer Research UK Gurdon Institute, Henry Wellcome Building of Cancer and Developmental Biology, Cambridge, UK    [2]Wellcome - MRC Cambridge Stem Cell Institute, Jeffrey Cheah Biomedical Centre, Cambridge Biomedical Campus, Cambridge, UK    [3]Physiology, Development and Neuroscience Department, University of Cambridge, Cambridge, UK    [4]Centre for Trophoblast Research, University of Cambridge, Cambridge, UK    [5]Division of Mammalian Embryology, Center for Stem Cell Biology and Regenerative Medicine, The Institute of Medical Science, The University of Tokyo, Tokyo, Japan    [6]Center for Genetic Analysis of Behavior, National Institute for Physiological Sciences, Okazaki, Japan    [7]Cancer Research UK Cambridge Institute, University of Cambridge, Li Ka Shing Centre, Cambridge, UK    [8]European Molecular Biology Laboratory, European Bioinformatics Institute, Wellcome Genome Campus, Cambridgeshire, UK    [9]Wellcome Sanger Institute, Wellcome Genome Campus, Cambridgeshire, UK

Correspondence: araa.venzor@gmail.com; a.surani@gurdon.cam.ac.uk
*Aracely Castillo-Venzor, Christopher A Penfold, and Michael D Morgan contributed equally to this work

regulator for PGC specification, as in humans and nonhuman primates, there is concomitant repression of SOX2, but not in mice, where Sox2 has a crucial role in PGC development (Campolo et al, 2013). The upstream regulators of the core PGC network, most prominently *SOX17*, remain to be fully deciphered in humans.

The site of human PGC specification also remains unclear. In cynomolgus and marmosets, PGCs are first observed in the amnion before gastrulation (Sasaki et al, 2016; Bergmann et al, 2022). At the later stages, PGCs are detected in the posterior epiblast, with the possibility of a dual origin (Sasaki et al, 2016; Kobayashi et al, 2017; Kobayashi & Surani, 2018). Note that in humans and nonhuman primates, the nascent amnion is among the first lineages to form from the epiblast (Xiang et al, 2019; Bergmann et al, 2022). PGC specification precedes amnion development in some nonprimate embryos, including bilaminar disc-forming species such as rabbits and pigs (Alberio et al, 2021; Kobayashi et al, 2021; Zhu et al, 2021). In pigs, PGCs arise from a pre-primitive streak (PS) and early PS stage-competent epiblast (Kobayashi et al, 2017), and in a rare Wk3 (Carnegie stage 7) human embryo, PGCs are associated with the primitive streak (Tyser et al, 2021). Here, we used our PSC-based model for PGC specification (Irie et al, 2015; Kobayashi et al, 2017) in conjunction with highly resolved single-cell transcriptome sequencing and integrative analysis with existing human and primate datasets to document the nature of the somatic components of the models and provide the context for PGCLC specification. We identified TFAP2A, generally considered an amnion marker (Shao et al, 2017a, 2017b) as thus far, the earliest and essential regulator of PGC fate. Loss of TFAP2A leads to an almost complete abrogation of PGCLCs in favour of a population of cells displaying SOX2 expression but has no significant effect on somatic lineages. Together, these observations extend knowledge of the PGC regulatory network and provide insights into the likely origin of human PGCs.

## Results

### A highly resolved transcriptional characterisation of PGC specification in embryoid bodies

Human PSCs in a primed state represent non-gastrulating post-implantation epiblast cells (Yu et al, 2021) with low competence for PGCLC fate (<5%) (Irie et al, 2015). PSCs can, however, acquire competence for PGC fate as self-renewing populations in media containing four inhibitors (henceforth called 4i conditions) (Gafni et al, 2013; Irie et al, 2015). Alternatively, PSC in response to WNT and activin signalling can also, albeit transiently, acquire competence for PGC-fate at ~12 h, known as pre-mesendodermal cells (henceforth called Pre-ME) (Kobayashi et al, 2017). Pre-ME progress to mesendoderm (ME) fate at 24 h when they lose competence for PGCLC specification and instead gain competence for definitive endoderm (DE; 60–80%) and mesoderm fates (70–90% efficiency).

The efficiency of PGCLC induction varies depending on the cell line (Chen et al, 2017), but we typically observe that between 10–40% of cells acquire PGCLC fate in the embryoid body (EB) specify; the remaining cells acquire somatic fates. Using our in vitro model (Kobayashi et al, 2017), we elucidate the transcriptional dynamics as

the Pre-ME cells undergo specification to PGCLCs in response to BMP (Fig 1A). We monitor changes in the transcriptional states by analysing the embryoid body using 10X Genomics single-cell RNA sequencing. We sampled EBs over a highly resolved time series between 12 h–96 h post-induction with additional comparative samples of conventional PSCs and PGC-competent populations (PreME). We also incorporated our earlier 4i cell samples (Tang et al, 2022) and d4 embryoids induced for PGCLC fate from 4i cells. Finally, we included flat cultures of DE and ME populations and Wk7 human gonadal PGCs for in vivo reference.

We first sought to establish the identity of detectable lineages using droplet single-cell RNA sequencing in the embryoid bodies and related controls (our collective dataset) which fell into 15 preliminary clusters (Fig S1A). Pseudo-bulk correlation of these clusters with existing 10X datasets showed a high degree of correlation between these clusters and amnion-like cells (AMLC), PGCLCs or mesoderm-like cells (MeLCs) (Zheng et al, 2019) and a comparatively low correlation with human syncytiotrophoblast and extravillous trophoblast (Vento-Tormo et al, 2018) (Fig S1B). Pseudo-bulk comparison with human in vitro embryo cultures (Xiang et al, 2019) and an in vivo CS7 human gastrula (Tyser et al, 2021), both Smart-seq2 samples, corroborate these observations with a higher degree of correlation between EBs and embryonic disc (EmDisc) or amnion but a substantially reduced correlation with other extra-embryonic tissues such as trophoblast and yolk sac mesoderm and preimplantation lineages (Fig S1C). Together these results suggest that in response to BMP, EBs progress to lineages of the peri-gastrulation embryo but not the extraembryonic tissues aside from the amnion.

Although pseudobulk comparisons help eliminate obviously absent cell types, such as trophoblast, it is not sufficient to fully resolve the cell lineages. In particular, correlation-based approaches were not adept at disentangling the relationship between similar cell types, particularly when comparing across sequencing platforms, for example, when comparing ESCs with the CS5 or CS6 stage EmDisc, which showed comparable correlation, with CS6 EmDisc incorrectly showing slightly higher correlation (Fig S1C). To gain further insights, we made use of the single-cell nature of our dataset and aligned our samples to a comprehensive range of existing embryonic datasets to refine cell annotations and create a human primate gastrulation and PGC atlas (Fig 1B). We included embryonic and amniotic lineages from human and cynomolgus in vitro-cultured embryos (Ma et al, 2019; Xiang et al, 2019; Zhou et al, 2019), in vivo human and marmoset gastrula (Tyser et al, 2021; Bergmann et al, 2022), and human gonadal PGCs (Guo et al, 2015; Li et al, 2017). We also include three in vitro models of human PGCLC induction based on the microfluidic models (Zheng et al, 2019), micropatterned gastruloids (Minn et al, 2020), and embryoid bodies from two other cell lines (Chen et al, 2019) (Table S1). We represent our aligned dataset in a 2D UMAP projection in Fig 1C, with cells coloured by sampling time. For comparison, we also provide aligned samples from the aligned human gastrula dataset atop our samples (Fig 1D), with the remaining datasets shown in Fig S1D–J. The initial assessment suggests a low number of doublets throughout with no apparent bias in any of the samples (Fig S1K). Clustering across all the datasets grouped cells into ~19 clusters, with the key clusters visualised for our data in Fig 1E. These clusters

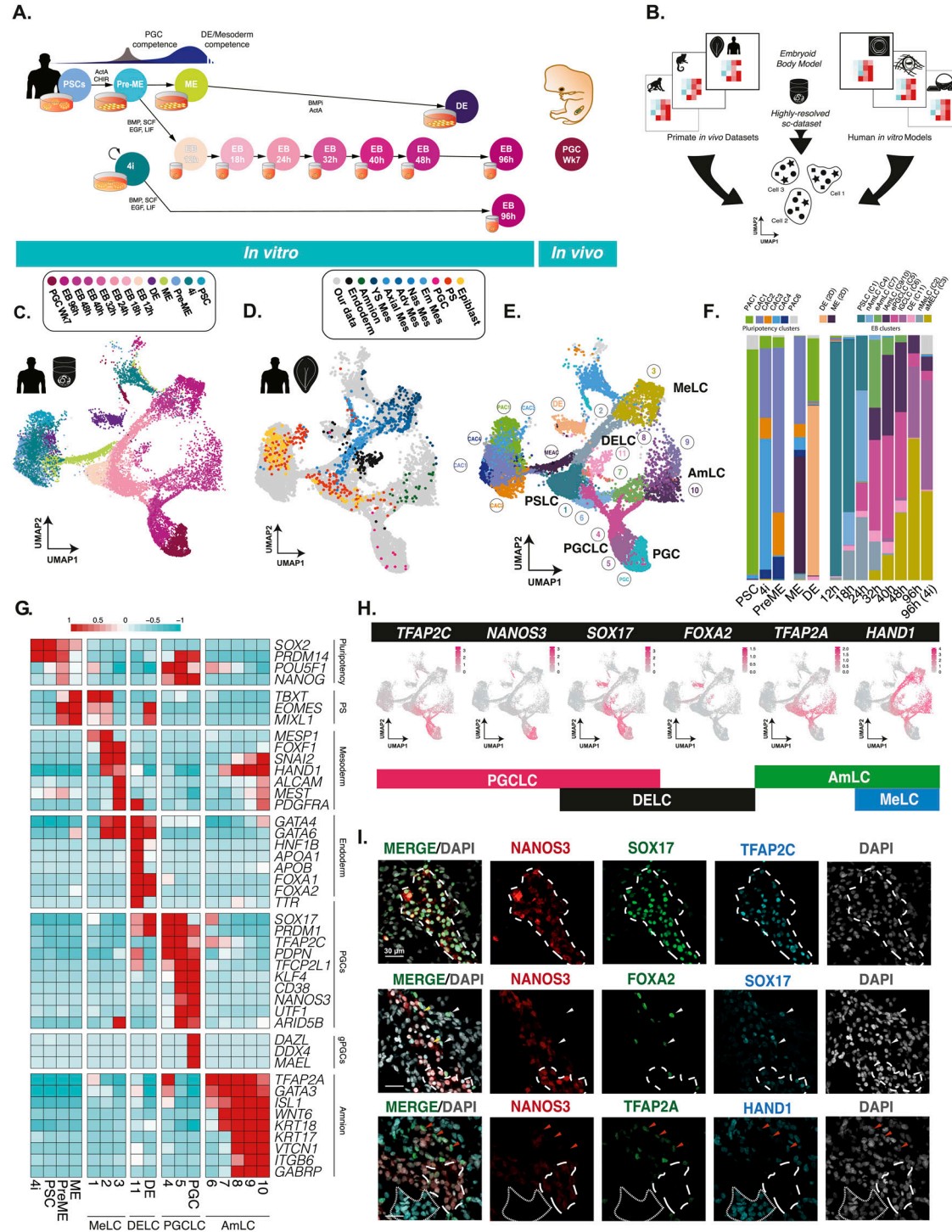

**Figure 1. A highly resolved roadmap of PGC development and gastrulation.**
**(A)** Experimental design for highly resolved RNA sequencing (10X) of our established PGCLC model, alongside PGCLC-competent populations, and in vivo and in vitro reference cell types. PSC, pluripotent stem cells; PreME, pre-mesendoderm (transient PGCLC-competent cells); ME, mesendoderm; 4i, four-inhibitor, self-renewing PGCLC-competent cells; DE, definitive endoderm; PGC, week 7 human gonadal PGCs; EB, embryoid body. **(B)** A schematic for the integration of our data alongside other human in vitro models and primate gastrulation datasets used to generate a roadmap of PGC and early human development. **(C)** Integrated data representation of our samples as a UMAP projection highlighted by collection time and sample type. **(D)** Integrated representation of the aligned human CS7 gastrula data, highlighted by cell type plotted atop our data (in grey). **(E)** Louvain clustering of the integrated dataset identified 19 clusters and highlighted four key terminal lineages. **(F)** Quantification of the composition of transient and terminal lineages associated with individual samples. **(G)** Heatmaps of pseudo-bulk expression for key markers show that the embryoid body diversifies into mesoderm-like cells (MeLCs), definitive endoderm-like cells (DELCs), primordial germ cell-like cells (PGCLC), and amnion-like cells (AmLCs). **(H)** A minimal combination of key expression markers can be used to identify cell fates, with *TFAP2C*⁺/*NANOS3*⁺/*SOX17*⁺ representing PGCLCs; *SOX17*⁺/*FOXA2*⁺ endoderm fate,

included several that were associated primarily with the embryoid bodies, and several more primarily associated with the pluripotent samples (labelled as pluripotency-associated cluster PAC1) or competent precursor populations (labelled as competence-associated clusters CAC1-4). The composition of EBs in terms of these clusters is shown in Fig 1F and highlights a rapid shift from pluripotency associate clusters to four distinct lineages, although a small but non-negligible population of cells associated with PSCs persisted in 4i-derived EBs.

A heatmap of the gene expression of relevant lineage markers confirms the presence of a transitory primitive streak-like population (PSLC; cluster 1), and MeLCs (cluster 2–3), definitive endoderm-like cells (DELCs) (cluster 11), AmLCs (cluster 6–10), and PGCLCs (cluster 4–5) within the EBs (Fig 1G). A limited number of cells fell into other clusters, although their numbers were typically low (1–30). In Fig S2A–I, we show critical differentially expressed transcription factor hubs during the formation of individual cell types. To visualise expression heterogeneity, we depict gene expression of six key lineage markers that, in combination, can be used to identify the cell fates in the EB (Fig 1H); expression of these genes was also confirmed at the protein level by immunofluorescence (IF) staining (Fig 1I). Notably, our detailed, integrated roadmap and characterisation show that at early stages, embryoid bodies contain subpopulations with molecular signatures similar to the PS, with cells at later time points showing transcriptional profiles associated with embryonic somatic fates (mesoderm and endoderm), and PGCLCs and amnion.

### Detection of PGC-competent population

Currently, there is no clear indication of what constitutes a PGC-competent population. To address this, we investigated how the precursor PreME cells gain competence for PGC fate. We also analysed the PGC-competent 4i cells against the noncompetent populations (PSCs and ME) (Tang et al, 2022). Comparisons with existing datasets suggested that our PSCs are transcriptionally like other PSCs (Chen et al, 2019; Zheng et al, 2019; Minn et al, 2020) (Fig S1D–F), aligning with a subset of cells from in vitro cultured human embryos labelled as EmDisc (Xiang et al, 2019; Zhou et al, 2019) (Fig S1G and H) and in vivo postimplantation epiblast (Tyser et al, 2021) (Fig 2A and B). Conversely, PreME cells cluster with pluripotent EmDisc samples (Xiang et al, 2019; Tyser et al, 2021) and cells labelled as epiblast, primitive streak, and mesoderm in a human CS7 gastrula (Tyser et al, 2021) (Fig 2C).

Quantification of the fraction of cell types in each of the competent or refractory samples suggests that PSCs were relatively homogenous in that most cells fell within a single cluster, whereas PreME, 4i, and ME fell into several (Figs 2D and S3A). Within these clusters, we note a PSC-dominant (pluripotency-associated cluster 1; PAC1), a ME-associated cluster (MEAC), a 4i-dominated cluster (CAC3). Three other subpopulations (CAC1, CAC2, and CAC4) were comprised of a mix of 4i and PreME samples, with CAC1 also containing ME samples. A handful of other smaller clusters were

also found, although these were typically small populations with <10 cells and grouped into CAC5.

Because distances in UMAP remain difficult to interpret (Chari et al, 2022 Preprint), we also visualise cells using diffusion maps (DM) to gauge the behaviour of these precursor populations compared with noncompetent PSCs, ME populations, and terminal-stage PGCLCs (Fig 2E and F). Within DM representations, these populations exist as a continuum of transcriptional states extending from PSCs to ME along diffusion components 2 (DC2) and DC3, with PGCLCs extending out along DC1 (Fig 2G).

Pairwise differential expression analyses between (competent) CAC1 and (noncompetent) PAC1 identified several likely regulators of competence, including EOMES, which has an identified role in PGC-competence (Chen et al, 2017; Kojima et al, 2021) and mesodermal markers SP5 and MIXL1 (Fig 2H). Additional pairwise comparison of the other competent-enriched subpopulations, for example, CAC2 and CAC4, against PAC1, identified similar markers, including OTX2, SOX11, TERF1, TCF7L1, SALL2, LIN28A, and TET1 (Fig S3B). Comparison of competent clusters against ME-associated cluster showed further up-regulation of mesoderm-related genes, MIXL1, GATA6, GSC, MESP1, ZIC2, and EOMES in ME-dominated cluster and concomitant reduction of pluripotency factor expressions (SOX2, SOX3, NANOG) and MYC in PGC-competent clusters (CAC1, CAC2, CAC4) in line with our previous findings (Tang et al, 2022) (Fig S3B).

A systematic comparison of ligand–receptor pair enrichment across clusters was made using CellPhoneDB (Garcia-Alonso et al, 2021). To simplify interpretations, we first subset by condition (PSC, PreMe, 4i) and only included interactions for PAC1 and CAC1–4. These results highlighted divergent NOTCH, FGF, and WNT signalling in competent populations versus primed (Fig S3C). CAC1, CAC2, and CAC4 appeared to be the dominant source of FGF2 in PreME with CAC3 also providing FGF3/4. In PSC populations, PAC1 formed a self-interacting source of FGF2, but far fewer FGF2-centric interactions were observed in 4i, with CAC2/3 instead appearing to act as a source for FGF1. Because of differences in the relative proportions of individual clusters in the PSC/4i/PreME samples, this suggests a general decrease in FGF2/3/4-related interactions in PreME and 4i populations versus PSC, an observation that correlates with previous studies in humans (Penfold et al, 2018 Preprint) and rabbits (Kobayashi et al, 2021) suggesting an antagonistic for FGF in PGC competence, with additional roles in PSC proliferation (Kobayashi et al, 2021) and ablation of amnion formation (Munger et al, 2022).

PAC1, CAC2, CAC3, and CAC4 were collectively a source of SFRP1/2 in PreME, with PAC1 and CAC1 a source of SFRP–WNT interactions in 4i. No enrichment of SFRP–WNT interactions was seen in the PSC population. In addition, we note strong divergence in DLK-NOTCH signalling in competent populations compared with primed populations, with competent populations showing minor changes in BMP–BMPR interactions compared with PSC populations, and 4i additionally displaying enrichment for CER1-related interactions. Together these analyses have identified molecular signatures that

---

TFAP2A⁻/HAND1⁺ mesoderm, and TFAP2A⁺/HAND1⁻ amnion fates. **(I)** Immunofluorescence of d4 EBs confirms expression patterns at the protein level, and the identity of PGCLCs, MeLCs, AmLCs, and DELCs.

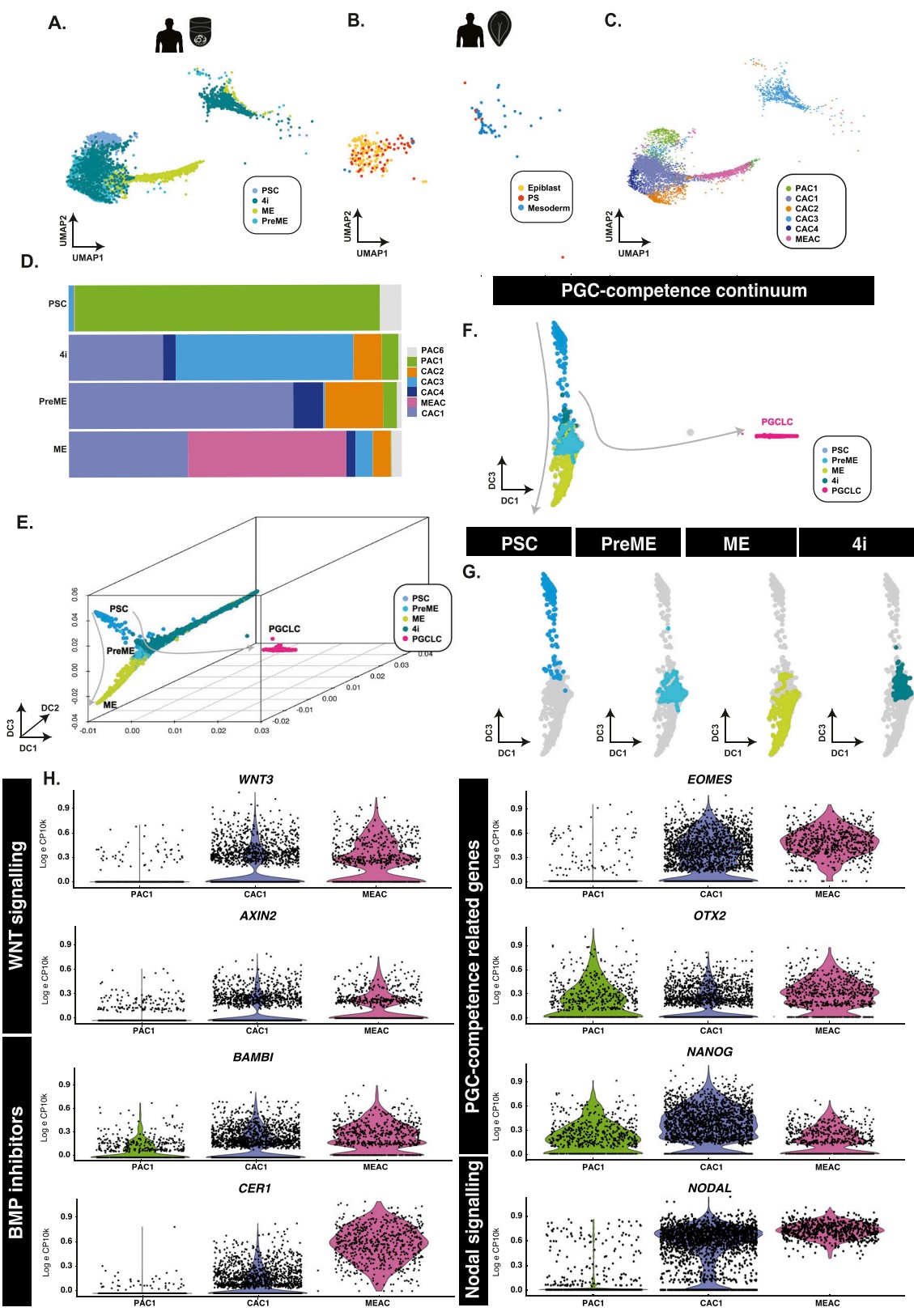

**Figure 2. PGCLC-competent populations form a continuum of states.**
**(A, B)** Aligned UMAP representations of pluripotent and PGCLC-competent populations, alongside (B) human in vivo samples show that PSCs align best to pluripotent epiblast cells, whereas competent (PreME and 4i) align to both epiblast-like and primitive streak-like populations. **(C, D)** Clustering of competent and non-competent cells and quantification (D) identified six main populations that identify a PSC-associate cluster (PAC1) predominantly associated with the PSC samples, a mesendoderm-

may underlie the transition from primed pluripotency to a competent state for PGC fate.

## Specification of PGCLCs in EBs represents a primitive streak-like stage

Based on the expression of marker genes, EBs first transition through a primitive streak-like stage before diversifying into mesoderm-like (MeLC), definitive-endoderm-like (DELC), and PGC-like states, with the additional formation of amnion-like cells, but with a notable lack of neural ectoderm populations (Fig S1D–J). Strikingly, these are lineages expected to arise at the posterior end of the developing embryo around the time of gastrulation.

To test this hypothesis further, we sought to map cells found in vitro to existing spatial transcriptomic datasets. Although spatially resolved human gastruloid datasets exist (Moris et al, 2020), these models capture the onset of somitogenesis (CS9). They are, therefore, more developmentally advanced than our model, which aligns well with data from CS5–7 embryos relevant to the emergence of PGCLCs. In this regard, we note recent comprehensive spatially resolved transcriptional datasets of marmoset embryos at CS5 and CS6 (Bergmann et al, 2022), where the peri-implantation development strongly resembles that of human embryo development at the morphological and transcriptional levels (Bergmann et al, 2022), including the conserved expression of *SOX17*, *PRDM1*, *TFAP2C*, and *NANOS3* in PGCs. Notwithstanding the differences in human and marmoset development timing, archival embryo collections allow consistent staging between species based upon Carnegie staging (Bullen et al, 1997; O'Rahilly & Müller, 2010).

To evaluate possible anterior–posterior bias, we mapped cells from our in vitro model to an existing 3D spatially resolved depiction of a CS6 marmoset embryo in which laser capture microdissection was used to generate a 3D spatially resolved transcriptome (see the Materials and Methods section). Together, they capture the critical cell types for comparison (Fig 3A) with gene expression patterns of critical markers shown in Fig 3B. Based on KNN spatial mapping (Fig 3C), we found that pluripotent stem cell populations mapped best to the anterior compartment (Fig 3D), in agreement with earlier studies (Tyser et al, 2021; Bergmann et al, 2022), although we could not rule out that these cells might have a better mapping to earlier stages, for example, CS4 bilaminar disc embryo, because no data for this stage are available. We found that the PreME population shifted towards the posterior end of the embryo, with amnion-like cells primarily mapping to the posterior amnion (Fig 3D). The basal cluster represents the 12 h embryoid body mapped to the posterior end of the EmDisc to a region expression *TBXT* and other primitive streak markers. Other cell lineages, including PGCLCs, showed an even stronger bias to the posterior end of the embryo, with PGCLC mapping to a distinct *NANOS3*-expressing region between the posterior-most EmDisc and amnion (Fig 3D). Together, these results provide further evidence that our model represents the development of the posterior

end of the embryo during gastrulation and suggests ongoing specification of both amnion and PGCs.

## Highly resolved time series reveal dynamics of cell trajectories

Having established the identity and spatial correspondence of key lineages, we next investigated the dynamics of individual cell fate decisions within the EB. We performed a label transfer from the human CS7 gastrula dataset (Tyser et al, 2021) to our data and separated EBs by collection time to visualise the emergence of cell types (Fig 4A). Twelve hours after inductive BMP cues, cells aligned primarily to the primitive streak (PS) with a limited pool of epiblast-like cells. Primitive streak-like cells (PSLCs) persisted in limited numbers until ~24–32 h. They showed sustained expression of *NODAL* (Fig S4). A small population of nascent and emergent mesoderm-like cells (denoted nMeLC and eMeLC, respectively) were observed in the competent populations (PreME) and appeared as early as 12 h in the EB, becoming more pronounced by 18 h, with these lineages roughly comprising cluster 2. The earliest PGCLCs arose around the 18 h mark, with amnion-like cells and definitive endoderm-like cells arising around 24 h.

We visualised the segregation of early mesoderm from precursors with a primitive streak-like identity using a diffusion map (Fig 4B). Cells not committed to mesoderm fate were instead predominately directed towards PGCLC or AmLCs. Both UMAP and DM representations suggest that PGCLCs and AmLCs stem from highly similar progenitor cells (Fig 4C). Interestingly, there remains some association between the PGCLC and AmLC branches until around 48 h, with a number of cells falling between the two main branches. Visualisation of the PGCLC branch alongside samples from the CS7 human gastrula shows an overlap between the gastrula samples and our Wk7 in vivo PGCs and late in vitro PGCLCs (Fig 4C). It is also worth noting that four other cells, initially labelled as PS in the human gastrula dataset, were also found to align to early PGCLCs and were reannotated accordingly. Together, these observations strongly suggest that the CS7 gastrula contains samples of PGCs at different stages of specification and that our in vitro model captures the dynamics of this early developmental trajectory at a much finer resolution. Cross comparison of CS7 PSCs with PGCLCs from various other in vitro models confirms a robust and conserved program of PGCLC specification centered around the *SOX17*/*TFAP2C*/*PRDM1* network with consistent up-regulation of *TFAP2A* and other genes (Fig S5A–D).

To identify potential signalling differences, we quantified interactions at different stages using CellPhoneDB. Fig S6A and B indicate targeted ligand–receptor interactions in PSLC, AmLC, and PGCLC populations. At 12 and 18 h, the PSLC populations appeared to have relatively little *BMP–BMPR*-related interactions but were targets of *NODAL* signalling via self-interactions and DE populations. The early n/eMeLC populations also served as a source of *CER1–WNT*-related interactions, with both early mesoderm-like and PSLCs potentially signalling via FGF ligands. In early time points,

---

associated cluster primarily found within ME samples, and several overlapping putative competence-associated clusters (CAC1, 2, 3, and 4) found mainly in either 4i or PreME samples. **(E, F, G)** In 3D (E) and 2D (F) diffusion map representations, samples sit along a continuum of overlapping states with competent populations predominantly in the middle along DC3 (G). **(H)** Violin plots of putative competence genes related to WNT and BMP signalling reveal a heterogeneous signalling response.

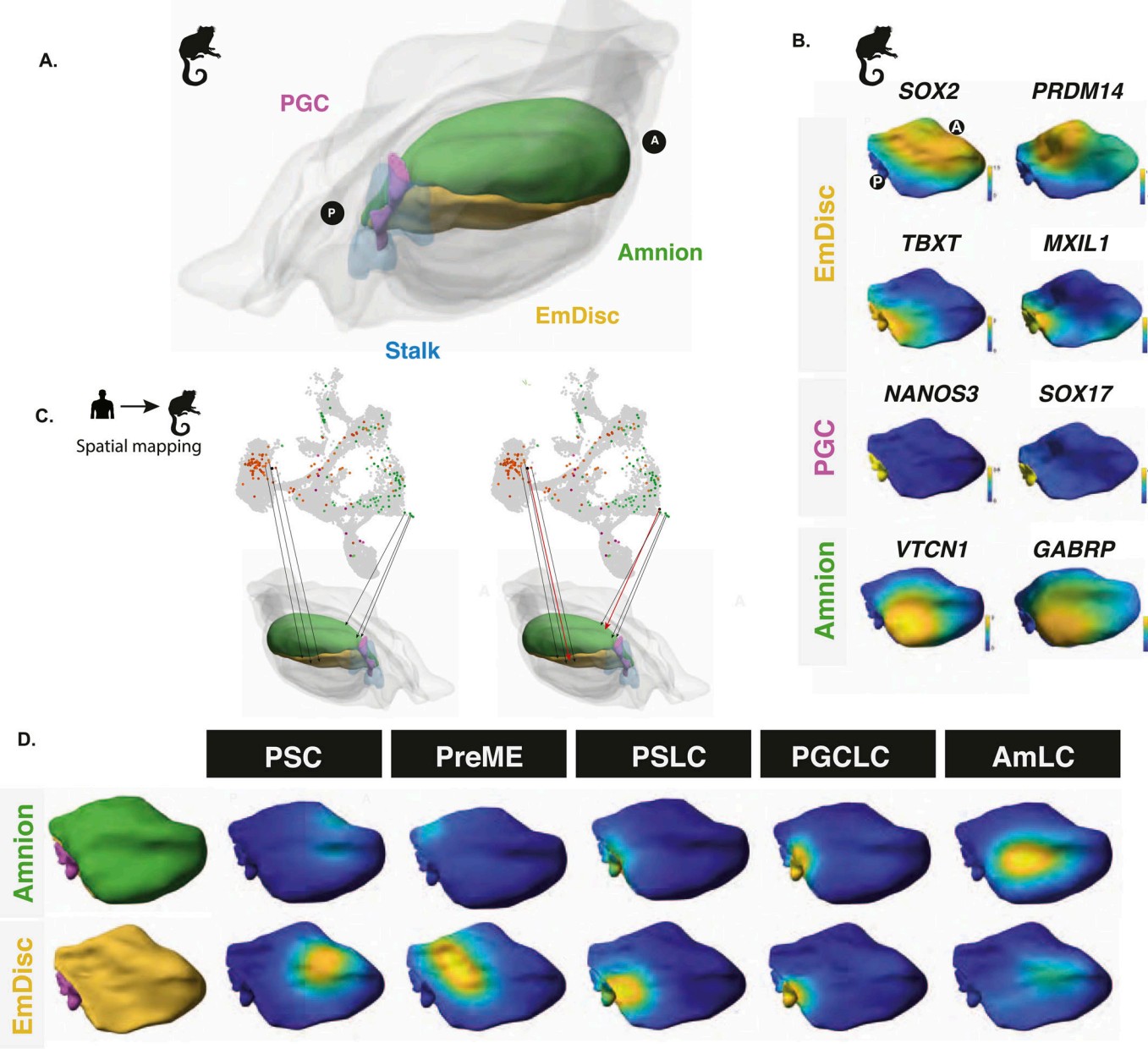

**Figure 3. Spatial mapping of embryoid bodies to gastrulating marmoset embryos reveals a posterior bias.**
**(A)** Spatially resolved marmoset embryos at CS6 with the embryonic disc in yellow, amnion in green, PGCs in pink, and stalk in blue. Extraembryonic tissues are depicted in grey. **(B)** Expression analysis in the marmoset reference embryo at Carnegie Stage 6 shows that the anterior embryonic disc is SOX2 positive and posterior regions are *TBXT* positive. Specified PGCs show similar expression patterns to humans, with *SOX17*/*NANOS3* expression, and amnion showing partial *GABRP*/*VTCN1* expression. **(C)** After the alignment of datasets visualised here as a UMAP, mapping of human in vitro cells to the marmoset reference embryo was achieved using KNN-based methods in PC space, with PSCs mapping best to the anterior embryonic disc. **(D)** Competent populations show a distinct posterior bias, with PGCLCs showing strong localisation to the posterior-most marmoset PGC region and AmLCs mapping to the amnion.

clear differences between AmLC and PGCLC populations emerged, with AmLC displaying more BMP-related signalling enrichment. These differences persisted to 96 h in PreME-derived PGCs, and enriched BMP responsiveness was observed in 4i-derived populations. AmLC and PGCLC also saw differences in *DLK–NOTCH* and *LEFTY–TDGF1*-related interactions. These observations highlight the divergent role of signalling cell types in EBs and the vital role of early lineages in shaping the balance of fates.

We quantified the dynamics of individual bifurcations by inferring lineage trajectories with Waddington-OT (Schiebinger et al, 2019), an optimal transport-based approach that allowed us to infer progenitor–progeny relationships between groups of cells statistically. By integrating these results with reduced dimensional representations of our time-course data, such as UMAP, DM or PCA, we sought to identify the most likely earliest progenitors of PGC specification in our data. Using the ancestor–progeny relationships

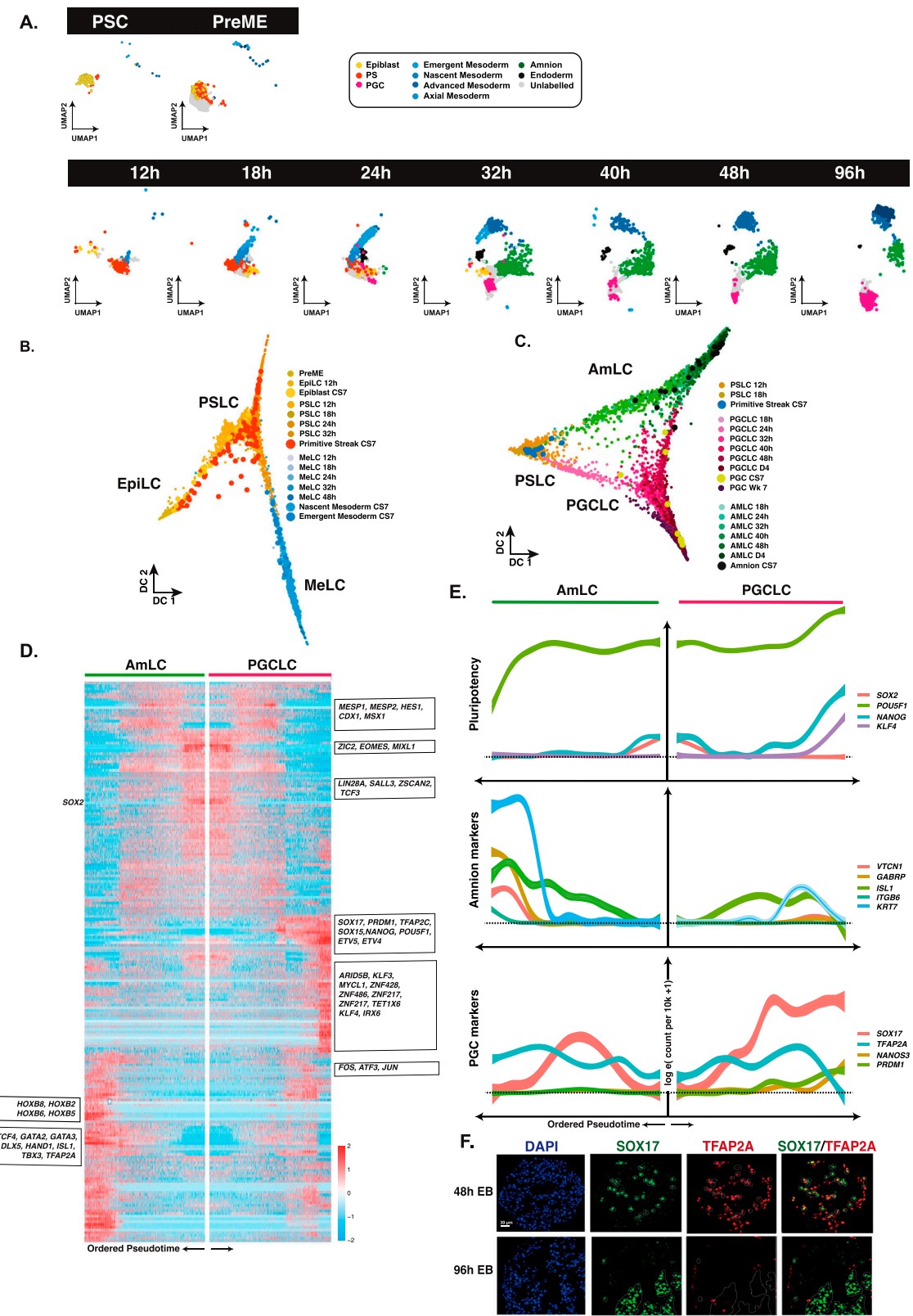

**Figure 4. Resolving the dynamics of bifurcations in embryoid bodies.**
**(A)** Visualisation of our data separated by sample time with cells annotated by transfer of labels from the human CS7 gastrula (Tyser et al, 2021); PS, primitive streak; PGC, primordial germ cell. Label transfer suggests embryoid bodies (EBs) develop first through a primitive streak-like stage, with some residual or emergent epiblast-like cells, and early emergence of both mesoderm-like cells and primordial germ cell-like cells, followed by amnion-like cells. **(B)** Diffusion map representation of specific clusters

computed by WOT, we inferred the broader lineages by first constructing a sparse network of clusters (Fig S7A and B), which were further grouped using a community-detection algorithm (see the Materials and Methods section). We assigned the inferred lineage identities to the single cells in these groups using broad marker gene expression patterns. As an initial check, we overlaid these WOT-inferred lineages onto our UMAP in Fig S7C, which demonstrated a good agreement with our earlier annotation-based lineage assignments with a high degree of correlation to our earlier cluster and marker-based annotations (Fig S7D). Using the Waddington-OT inference, most terminal cell fates were effectively traced to 24 h, with some cell groups traced to earlier stages.

Early mesoderm populations progressed from a PS-like state through a nascent mesoderm-like state (nMeLC) expressing *MESP1/2* and *T* to an emergent mesoderm-like state (eMeLC), representing the highest levels of *MESP1/2* and down-regulation of *T* (Figs S1L and S7E and F). Between 24–32 h, a *PDGFRA*-positive population emerged, aligned to advanced mesoderm of the human gastrula (denoted advanced mesoderm-like cells; aMeLC), concomitant with the gradual loss of nMeLC and eMeLC subpopulations. By ordering gene expression along a diffusion pseudotime analysis, we observed the late up-regulation of several advanced mesoderm markers, *HAND1*, *SNAI2*, and *GATA6* (Fig S7E–G). As the earliest specified fate, nascent and emergent mesoderm cells show down-regulation of pluripotency factors *POU5F1*, *SOX2*, and *NANOG* (Fig S2A and B) and up-regulation of several genes that may influence the balance of fates within the embryoid body, including *BMP4*, *WNT5A*, and *CER1*, and extracellular matrix genes (see e.g., Fig S4).

From 24–32 h, a limited pool of *SOX17*-positive endoderm-like cells were bifurcated from the PS-like subpopulation and showed sustained *NODAL* expression with subsequent up-regulation of endoderm markers *FOXA1/2* (Fig S7H and I) and down-regulation of *POU5F1* (Fig S2E). Although the number of cells in this population appeared to be fewer than that of other cell lineages, it was nevertheless a conserved feature across in vitro models.

Around the 18-h mark, the earliest PGCLCs bifurcated from a progenitor population with strong up-regulation of *SOX17*, *TFAP2C*, and *PRDM1* (see e.g., Fig S2G and H) and subsequent expression of *NANOS3*. PGCLCs also showed up-regulation of *WNT2* with early PGCLCs expressing *NODAL* (see e.g., Fig S4).

Indeed, a comparison of PGCLC precursor cells in high and low PGC-competence cell lines (Chen et al, 2019) revealed *NODAL* to be differentially expressed, consistent with a recently observed role for *NODAL* in PGCLC specification (Jo et al, 2021 *Preprint*). Slightly later, at 24 h, an AmLC branch also became evident, expressing *TFAP2A* and, at later time points, *ISL1*, a LIM/homeodomain transcription factor protein recently identified as an amnion marker (Guo et al, 2020 *Preprint*; Yang et al, 2021) (Fig S2C and D). This AmLC branch shows an expression of *WNT6* (Fig S4). We identified

differentially expressed genes along the separate AmLC and PGCLC lineages using the diffusion pseudotime ordering of single cells (Fig 4D; see Supplementary Materials). Within these pseudotime trajectories, we observed that both AmLC and PGCLC showed early coordinated expressions of *EOMES*, *MIXL1*, and *ZIC*, together with rapid down-regulation of *SOX2*. Moreover, we observed late expression of *VTCN1*, *GATA3*, *GATA2*, *ISL1*, and *HAND1* in AmLCs, whereas the PGCLC trajectory showed late expression of PGC markers *SOX17*, *PRDM1*, *TFAP2C*, *SOX15*, *KLF4*, *LIN28*, and *POU5F1*. Fig 4E shows the divergent expression patterns of crucial TFs over pseudotime to trace their rise and fall to AmLC versus PGCLC trajectories. We note an initial up-regulation of *SOX17* in AmLC and PGCLCs that is transient in AmLC but sustained in PGCLCs. Surprisingly, *TFAP2A*, which is generally considered an amnion or trophoblast marker (Krendl et al, 2017; Zheng et al, 2019; Minn et al, 2020), precedes *SOX17* expression and is transiently co-expressed with *SOX17* in the PGCLC trajectory. Whereas AmLCs maintain *TFAP2A* expression, there is down-regulation in PGCLCs, which was confirmed by immunofluorescence staining at the protein level (Fig 4F). Staining of EBs for *TFAP2A* and *SOX17* confirmed their co-expression at early time points, whereas, in the 96 h EB, *TFAP2A* expression is exclusive to AmLC and *SOX17* to PGCLCs and DELCs. These results, taken together, highlight the complex dynamics of PGCLC specification within our model system and identify several putative markers of specification. The most interesting was the early and transient expression of *TFAP2A* in PGCLCs. TFAP2A is an early BMP response gene that shares the TF-binding site with TFAP2C (Krendl et al, 2017). Given that we previously found TFAP2 motifs around PGC-related genes (Tang et al, 2022), and that the TFAP2 family can play complementary roles, it is interesting to see if *TFAP2A* plays a role in PGCLC specification before the onset of *TFAP2C* expression.

## TFAP2A is the most upstream crucial regulator of PGC specification

The *TFAP2A* signature identified in our data suggests a potential role in regulating PGCLC fate. Indeed, *TFAP2A* was previously shown to be up-regulated in PGCLC cells over competent and non-competent PSC populations (Irie et al, 2015) and verified in a subsequent study (Chen et al, 2019), prompting the hypothesis that TFAP2A might regulate PGCLC formation. However, *TFAP2A* is one of several thousand genes expressed in PGCLCs, and neither study attempted to verify a role for TFAP2A in PGCLC fate functionally.

To determine if the transient *TFAP2A* expression has a role, we examined the consequences of loss of function by comparing the effect on PGCLC induction in the parental PSC line versus TFAP2A loss of function (Krendl et al, 2017) (Fig 5A). We observed a reduction in PGCLCs in *TFAP2A* mutant cells compared with parental controls by FACS using antibodies for PGC-surface markers PDPN and AP

reveals the bifurcation of mesoderm from the PS-like progenitors, with the remaining PS-like cells destined for other lineages. Because of the limited number of cells, samples from the CS7 human gastrula are depicted as larger data points to aid visualisation. **(C)** A diffusion map representation of AmLC and PGCLCs shows bifurcation from common progenitor populations, with a sustained association until 48 h. Superimposition of cells from the CS7 gastrula labelled as PS, amnion or PGCs shows an early alignment of human PGCs to PGCLCs. **(D)** A heatmap representing differentially expressed genes between AmLC and PGCLC ordered by pseudotime. **(E)** Line plot representations of essential genes ordered by pseudotime show early up-regulation of TFAP2A in both PGCLC and AmLCs, which is sustained in AmLC. **(F)** IF shows TFAP2A in early PGCLCs at 48 h (SOX17/TFAP2A double-positive) is lost by 96 h.

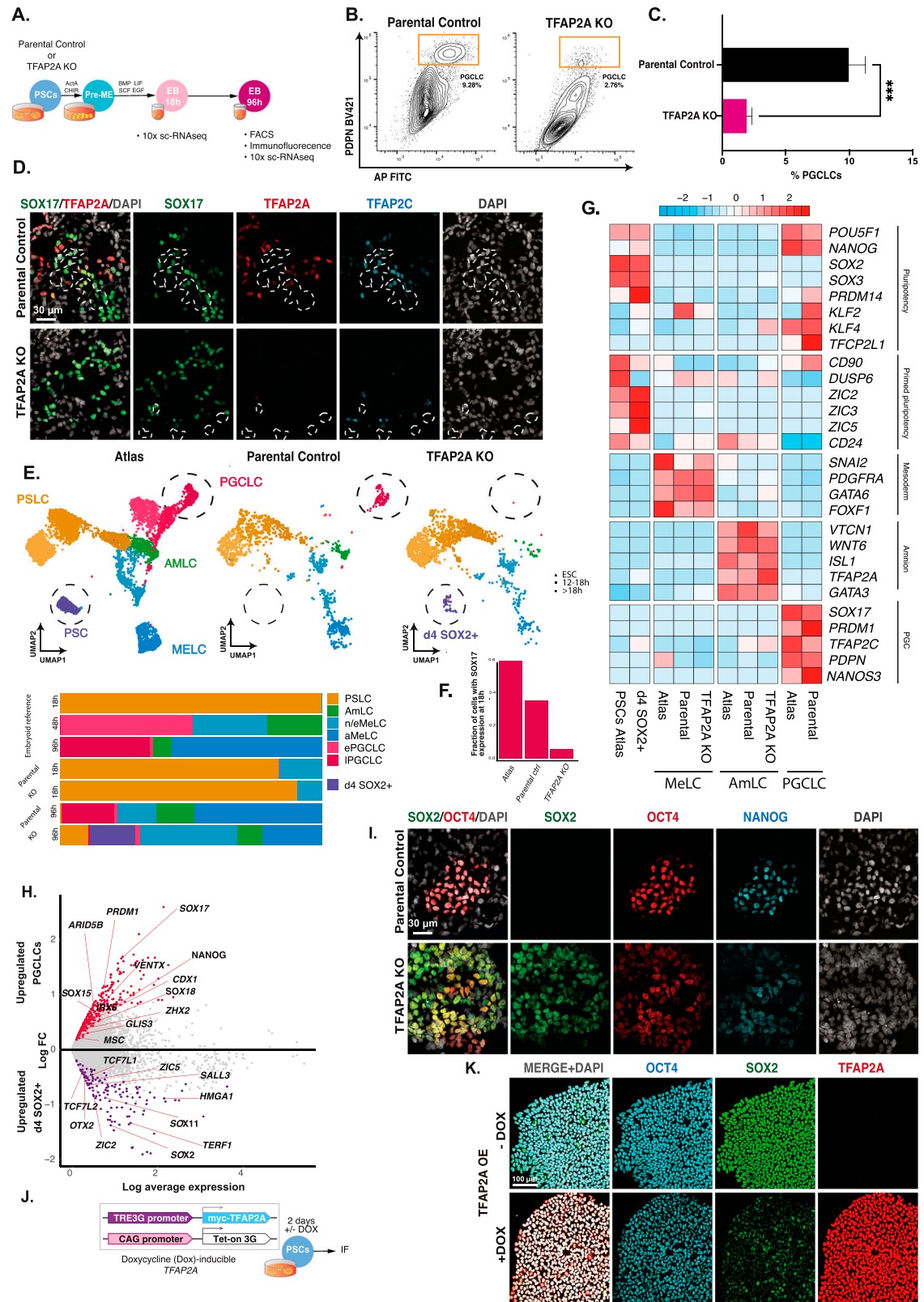

**Figure 5. TFAP2A is a regulator of PGCLC fate.**
**(A)** Experimental design for testing the role of TFAP2A in PGC specification using TFAP2A knockout line. **(B, C)** FACS plots and quantification based on IF-labelled PDPN expression reveal a decrease in the % of PGCLCs induced in TFAP2A KO EBs compared with WT parental control. **(D)** Immunofluorescence shows co-expression for SOX17, TFAP2A, and TFAP2C in d4 EB. **(E)** Aligned UMAPs and bar plot quantification for the reference atlas versus H9 parental control and H9 TFAP2A KO further corroborate the

(2.78% versus 9.28%) (Fig 5B). The result is a consistent and statistically significant reduction of PGCLC specification in TFAP2A KO EBs (Fig 5C), which was further validated by immunofluorescence staining of d4 EBs generated from TFAP2A knockout cells, compared with parental control (Fig 5D).

To characterise the phenotypic consequences of TFAP2A loss of function, we generated 10X scRNA-seq datasets for two time points: 18 h, just before the diversification of distinct lineages in embryoid bodies, and 96 h, when terminal cell fates have been established. We integrated these time points with our existing EB dataset containing all cell types for reference using Seurat. For this alignment, we generated a new clustering visualised on a UMAP in Fig 5E. In 18 h samples, both the TFAP2A KO and parental lines mapped best to PSLCs, although the number of SOX17-positive cells in the KO line was significantly reduced compared with parental line EBs (Fig 5F; P < 0.0001 FET).

As before, by 96 h, embryoid bodies from the parental line showed the precise formation of a MeLC expressing PDGFRA, amnion-expressing VTCN1, and PGCLCs expressing NANOS3 by 96 h (Fig 5F). Although there were no obvious detectable DELCs in either the parental or TFAP2A KO line by scRNA-seq, likely because of their limited cell numbers, immunofluorescence analysis shows rare SOX17, FOXA2 double-positive cells in the EB (Fig S7A). In contrast, in EBs from TFAP2A KO cells, the PGCLC lineage was virtually absent (Figs 5E and S8A and B; significantly decreased compared with parental control, P < 0.0001, FET). However, a MeLC and AmLC population was present, with a slight decrease in the fraction of MeLCs observed compared with parental lines and no statistically significant difference in the number of AmLCs. Although the TFAP2A KO appeared to lack PGCLCs, we observed a new subpopulation of cells at 96 h clustered alongside pluripotent cells (Fig 5E). This population, absent in the parental line and rare at the 18 h mark in the KO line, showed expression of SOX2 and other pluripotency markers (hereafter referred to as SOX2+d4 cells; Fig S8C).

To help establish the authenticity of the other fates, we generated a cross-correlation heatmap (Fig S8D). The SOX2+d4 cluster is most like PSCs in the reference population. AmLCs in the KO cluster were highly similar to AmLCs in the parental line and the reference line, with MeLCs, also showing consistency across all cell lines. Together these observations suggest no significant effect of TFAP2A loss of function for MeLCs or AmLCs specification. Immunofluorescence analysis confirmed the presence of AmLCs (GATA3+ HAND1+), MeLCs (HAND1+) (Fig S8E), and DELC cells (SOX17+, FOXA2+) in EBs with TFAP2A KO, except for a minimal number of PGCLCs (SOX17+, OCT4+) (Fig S8A). We, therefore, focused on PGCLCs and the SOX2+ population that we detected in EBs after TFAP2A KO.

Differential expression analysis of the SOX2+d4 population compared with parental-line PGCLCs showed that the SOX2+d4 cells expressed pluripotency and neural-plate factors, ZIC2, ZIC5, SOX11,

and OTX2. In contrast, PGCLCs showed the expression of germ cell markers SOX17, PRDM1, SOX15, ARID5B, TFCP2L1, and VENTX (Fig 5G). We found up-regulation of naïve markers of pluripotency and neuronal lineage-associated genes in the SOX2+d4 population compared with PSCs in the reference atlas; markers included PRDM14, KLF4, KLF6, and TFAP2C, and neuro-related genes ZIC2, ZNF292, FOXN3, POU3F1, SOX11, SOX4, ZIC5, and SALL3 (Fig S8F).

We performed immunofluorescence staining at d4 EBs to validate our findings and found expression of SOX2 in TFAP2A KO cells even after 4 d of cytokine exposure. SOX2+ d4 cells also showed the co-expression of OCT4 and NANOG (Fig 5H). By contrast, there is a rapid down-regulation of SOX2 upon BMP exposure (Fig S8C) in WT cells, which is critical for efficient PGCLC specification (Lin et al, 2014).

We investigated if TFAP2A can potentially target SOX2 for down-regulation using a stable dox-inducible TFAP2A PSC line (WIS2). Upon doxycycline induction for 2 d of TFAP2A in PSCs cultured in E8 medium (Fig 5I), we observed a substantial reduction in SOX2 levels. After TFAP2A overexpression by immunofluorescence (Fig 5J), POU5F1 was also slightly reduced. Together, our results suggest that TFAP2A is a regulator of PGCLC fate and may participate in the up-regulation of SOX17 and down-regulation of SOX2 and other targets impeding PGCLC specification (Fig S8G).

# Discussion

In vitro models have been crucial for unravelling the transcriptional network responsible for human germ cell competence and specification (Teo et al, 2011; Irie et al, 2015; Sasaki et al, 2015; Chen et al, 2017; Kobayashi et al, 2017; Kojima et al, 2017, 2021; Sybirna et al, 2020). In this study, we characterise in vitro models for the derivation of PGCLCs from PSCs by highly resolved single-cell transcriptomics and comprehensive comparison to in vivo references in human, nonhuman primates, and other in vitro models of gastrulation.

Notably, we found that PGC-competent PreME cells exist transiently within a continuum of states extending from PSCs to mesendoderm (ME). Our analysis showed that clusters enriched for PGC-competent populations present a particular signalling signature characterised by active Nodal and WNT signalling. There is low expression of BMP inhibitors (BAMBI and CER1) in competent cells compared with the ME-dominated cluster (Fig 2H), with the highest levels in non-competent cluster 22, which likely impedes PGC specification. BAMBI is a direct target of WNT signalling (Sekiya et al, 2004), whereas activation of CER1 occurs via both WNT and Nodal signalling (Katoh & Katoh, 2006; Martyn et al, 2019). PGCLC-competent clusters also show transient down-regulation of OTX2 and higher levels of NANOG compared with non-competent

drastic reduction in the numbers of PGCLCs in the knockout line. These results further suggest the emergence of a new SOX2+ population that occurs after 18 h and aligns with pluripotent stem cells in the reference atlas (D4 SOX2+). **(F)** Fraction of cells positive for SOX17 expression at the 18 h time point in reference atlas, parental control, and TFAP2A KO cells. **(G)** Row-normalised gene expression demonstrates consistent expression in AmLC and MeLC in the TFAP2A KO line. D4 SOX2+ cells show the expression of pluripotency genes. **(H)** Volcano plot for differentially expressed genes between the d4 SOX2+ cluster in TFAP2A KO versus PGCLCs in parental control. **(I)** Immunofluorescence of d4 parental EBs shows OCT4 NANOG double-positive cells (PGCLCs) but not in TFAP2A KO EBs; instead, there are OCT4, NANOG, and SOX2 triple-positive cells. **(J)** An inducible system to test the role of TFAP2A overexpression on SOX2 expression in PSCs. **(K)** Immunofluorescence for OCT4, SOX2, TFAP2A in PSCs after TFAP2A induction.

clusters, which we recently found is conducive to transition to the PGCLC state (Tang et al, 2022). Concomitantly, there is an increase in the levels of *EOMES*, which has a prominent role in human PGC competence (Chen et al, 2017; Kojima et al, 2017), but further activation of mesoderm factors hinders PGC specification. The tight signalling axis, transcription factor levels, and intrinsic heterogeneity modulating competence are consistent with a relatively small number (~100–200) of founder PGCs in vivo (Saitou et al, 2002; Kobayashi et al, 2017).

Specification of PGCLCs in vitro occurs within a 3D aggregate of a hitherto poorly characterised fraction of somatic components. Currently, PGCLCs can be induced in various 2D aggregates but more efficiently in 3D embryoids, highlighting the importance of the structure, cell–cell interactions or signalling from adjacent tissues (Minn et al, 2020, 2021). Here, we have shown that these somatic cells collectively represent those in the posterior region of the embryo during gastrulation. Among the somatic cell types, we note the early formation of MeLCs, which display strong expression of *BMP*, *WNT*, and ECM components that may be important for PGC fate and potentially play a similar role to that of extraembryonic mesoderm in the embryo, and endoderm-like cells that are double positive for *FOXA2/SOX17*. Furthermore, we also observe the emergence of *ISL1/VTCN1*-expressing amnion cells, providing evidence that amnion formation continues from the posterior epiblast during gastrulation, as recently suggested in a study on marmoset (Bergmann et al, 2022).

Mapping the cells to a 3D primate embryo showed that PSCs best correspond to the anterior region of the EmDisc, whereas PreME cells shifted towards the posterior end. Conversely, cells within the newly formed embryoid body at 12 h, which transcriptionally resemble a primitive streak, mapped best to the posterior end of the EmDisc, with PGCLCs mapping to a *SOX17/TFAP2C/NANOS3*-positive region at the boundary between the posterior-most epiblast and amnion.

The origin of human PGCs remains to be fully resolved because of the inaccessibility of human embryos, but bilaminar disc embryos from other species provide valuable information. In species such as rabbits and pigs, PGCs originate from the posterior epiblast, but the amnion develops later, indicating that the development of the amnion and PGCs in some cases is temporally unconnected. In humans and nonhuman primates, development of the amnion commences before PGC specification, but according to our work and by others (Bergmann et al, 2022; Rostovskaya et al, 2022) amniotic cells continue to emerge later from the posterior epiblast, coincidentally with the specification of PGCs at the time of primitive streak formation. In cynomolgus monkeys, the earliest PGCs have been reported in the amnion, with most found later in the epiblast. One possibility is that these early PGCs may arise from intermediate cells that are en route to amnion fate but are not yet fully committed squamous amniotic epithelium, as suggested by our DMs (Fig 4C). To contribute to the founder PGC pool, PGCs arising in the amnion would need to migrate against the continuing amnion growth. We posit that at this stage of development in humans and nonhuman primates, amnion cells continue to be specified with nascent PGCs arising at the posterior-most end of the epiblast during the early PS stage.

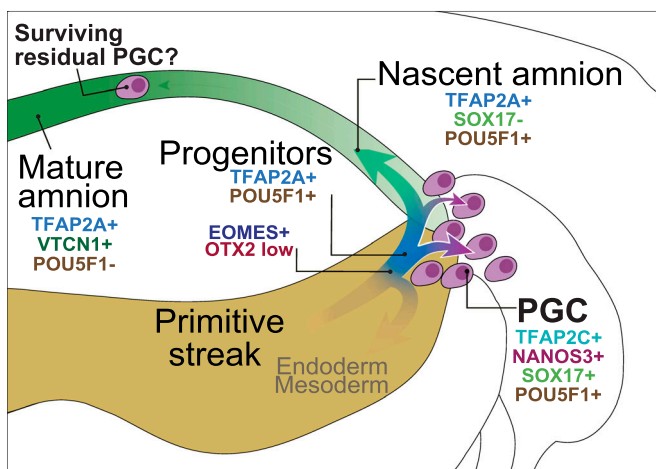

**Figure 6.   A schematic for the origin of PGCs in humans.**
PGCs are specified from a population of *TFAP2A*-positive progenitors at the posterior end of the embryonic disc. PGCs in the amnion specified at an earlier stage might contribute to the founder PGC pool if they can migrate against the flow of nascent amnion expansion.

In our model, AmLC and PGCLC progenitors display early expression of *TFAP2A*, a pioneer factor previously suggested to be associated with the amnion (Shao et al, 2017a, 2017b). Although there is subsequent down-regulation of *TFAP2A* in PGCLCs, the expression is sustained in the amnion. Surprisingly, the knockout of TFAP2A did not have a detectable effect on AmLCs, which merits further investigation but notably resulted in an almost complete abrogation of PGCLCs.

In PGCLCs, *TFAP2A* is rapidly replaced by the expression of *TFAP2C*, suggesting otherwise mutually exclusive expression after a brief window of co-expression. Interestingly, TFAP2A shares the same transcription factor-binding motif as TFAP2C (Krendl et al, 2017). TFAP2C is essential for PGC development (Kojima et al, 2017). It acts as both an activator and a repressor during PGC specification, but it is insufficient for PGC fate in the absence of cytokines (Kobayashi et al, 2017). In the PGCLC pseudotime trajectories, we saw early up-regulation of *TFAP2A* (12 h), followed by expression of *SOX17* and *TFAP2C* (18 h), and later, activation of *PRDM1* (24–32 h) by SOX17 (Tang et al, 2022) (Fig S9). In some instances, TFAP2A functions similarly to TFAP2C (Hoffman et al, 2007; Li & Cornell, 2007). Our work suggests that TFAP2A expression is transient but essential for initiating the PGC transcriptional network and may directly or indirectly repress SOX2 and other factors.

TFAP2A KO EBs show an emergent population (SOX2+ d4 cells) found to align to pluripotent stem cells, with the expression of the core pluripotency genes; *SOX2*, *POU5F1*, and *NANOG*. Differential gene expression between PSCs and SOX2+ d4 cells shows aberrant up-regulation of naïve markers *KLF4*, *TFAP2C*, and *PRDM14* and genes associated with the neuronal lineage, including *ZNF292*, *FOXN3*, *SALL3*, *ZIC2*, *POU3F1* in SOX2+ d4 cells.

There is rapid down-regulation of SOX2 during human PGCLC induction (Kobayashi et al, 2017); indeed, sustained SOX2 expression prevents PGCLC specification because of elevated differentiation into the neuronal lineage (Lin et al, 2014), which could, in part, explain the expression of related neuronal markers in the TFAP2A

mutant cells. The combinatorial role of SOX17–OCT4 involved in human germ cell fate (Tang et al, 2022) might benefit from the repression of SOX2 to favour the SOX17–OCT4 interaction on the compressed motif (Jostes et al, 2020). Interestingly, our preliminary findings indicate that TFAP2A may have a potential role in the up-regulation of SOX17 in PGCLC-competent cells (Pre-ME). Further studies are necessary to fully understand the relationship between TFAP2A and the establishment of PGCLC.

In summary, we provide insight into early human development with the transient emergence of the germ cell-competent PreME cells in a model mimicking human gastrulation starting with PSC. Our study suggests continuing emergence of the amnion from the posterior epiblast at the time of PGC specification during early gastrulation; the amnion and PGC likely arise from highly similar progenitors exemplified by TFAP2A expression. The loss of function has a marked effect on PGC specification but without a detectable effect on the amnion. Accordingly, PGCs likely emerge from the posterior epiblast predominantly, notwithstanding a subset in the early amnion, as summarised in Fig 6. Of great interest would be to test, when possible, the predictions we make by direct observations in extended cultures of developing human embryos.

## Materials and Methods

### Cell culture

In the study, the H1 NANOS3-tdTomato PSC line, which was previously generated in the laboratory (Kobayashi et al, 2017), and H9 parental and TFAP2A KO cells, kindly provided by Micha Drukker (Krendl et al, 2017) were both used. All cell lines were confirmed to be free of mycoplasma. The PSCs were cultured on plates coated with vitronectin (Thermo Fisher Scientific) in Essential 8 medium (Thermo Fisher Scientific) as per the manufacturer's instructions. The medium was changed daily, and cells were incubated at 37°C in 5% CO2 and atmospheric O2. The cells were passaged in a ratio of one to five every 3–4 d using 0.5 mM EDTA (Sigma-Aldrich) in PBS (Thermo Fisher Scientific) without breaking the cell clumps. After thawing a new vial, cells were only kept in culture for 8–10 passages before thawing a new vial.

For the 4i condition, undifferentiated PSC cells were grown on irradiated mouse embryonic fibroblasts (MEFs) (GlobalStem) in a 4i medium (Irie et al, 2015). These cells were passaged every 3–5 d using TrypLE Express (Gibco), quenched with MEF media, and then filtered with a 50-$\mu$m cell filter (PERTEC). The culture was supplemented with 10 $\mu$M of the ROCK inhibitor Y-27632 (TOCRIS Bioscience) to prevent differentiation for 24 h after passage.

The mesendoderm induction was carried out, as previously reported by Kobayashi et al (2017). The PSCs were dissociated into individual cells using TrypLE Express and then seeded onto vitronectin-coated plates at a density of 500,000 cells per well in a six-well plate. The cells were cultured in a mesendoderm-induction medium for 10–12 h. The ME medium used in this study is based on aRB27.

The process of inducing PGCs, mesendoderm, PGCLCs, and definitive endoderm from NANOS3-tdTomato reporter PSCs was carried

out as described in Kobayashi et al (2017) using the aRB27 basal medium. This medium is composed of Advanced RPMI 1640 Medium (Thermo Fisher Scientific) supplemented with 1% B27 supplement (Thermo Fisher Scientific), 0.1 mM non-essential amino acids (NEAA; Thermo Fisher Scientific), 100 U/ml penicillin, 0.1 mg/ml streptomycin (Thermo Fisher Scientific), and 2 mM L-glutamine (Thermo Fisher Scientific).

To induce PGCLCs, PreME cells were dissociated into single cells using TrypLE and seeded at a density of 4,000 cells per well into Corning Costar Ultra-Low attachment 96-well plates (Sigma-Aldrich) in human PGCLC induction medium. This medium is composed of the aRB27 medium supplemented with 500 ng/ml BMP4 (Department of Biochemistry, University of Cambridge), 10 ng/ml human LIF (Department of Biochemistry, University of Cambridge), 100 ng/ml SCF (R&D systems), 50 ng/ml EGF (R&D Systems), 10 $\mu$M ROCKi, and 0.25% (vol/vol) poly-vinyl alcohol (Sigma-Aldrich). Cells were cultured as floating aggregates for 2–4 d.

For mesendoderm induction, trypsinized human PSCs were seeded on vitronectin-coated dishes at a density of 200,000 cells per well in a 12-well plate and cultured in a mesendoderm-induction medium for 12 h (PreME) and 24 h (ME). The mesendoderm induction medium contained aRB27 medium supplemented with 100 ng/ml activin A (Department of Biochemistry, University of Cambridge), 3 $\mu$M GSK3i (Miltenyi Biotec), and 10 $\mu$M of ROCKi (Y-27632; Tocris bioscience). To induce definitive endoderm from ME, the mesendoderm induction medium was replaced with a definitive endoderm induction medium after washing with PBS (Thermo Fisher Scientific) once and cells were cultured for 2 d.

PSCs, PreME, ME, DE, and PGCLCs cells were harvested using a solution of 0.25% trypsin/EDTA (Life Technologies) at 37°C for a period of 5–15 min. The DE cells were then stained with an anti-CXCR4 antibody (PerCP-Cy5.5 conjugate; BioLegend) before being sorted using an SH800Z Cell Sorter (Sony) and analysed with FlowJo software.

### Collection of human PGCs from human embryos

The use of human embryonic tissue in this study was conducted under the approval of the NHS Research Ethical Committee in the UK, with a reference number of 96/085. The tissue samples were obtained from Addenbrooke's Hospital in Cambridge, with the full consent of patients who had undergone medical or surgical termination of pregnancy. The developmental stage of the human embryos was determined by measuring the crown-rump length and observing anatomical features such as limb and digit development, using the Carnegie staging system as a reference (CS). The sex of the embryos was determined using PCR-based techniques, as previously described in a study by Bryja & Konečný (2003).

### FACS

PSCs, 4i, PreME, and ME cells were harvested using TrypLE (GIBCO) at 37°C for 2–3 min. Embryoid bodies were collected and dissociated into individual cells using trypsin-EDTA solution (0.25%) (Life Technologies) at 37°C for 5–15 min. The dissociated cells were washed and resuspended in FACS buffer (PBS containing 3% FCS). The samples from the primitive endoderm (DE) were then stained

with PerCP-Cy5.5 conjugated anti-CXCR4 antibody (BioLegend) at 4°C for 1 h. After washing with PBS (Thermo Fisher Scientific), the samples were stained with DAPI and sorted on a SONY SH800 sorter. Flow cytometry data were analysed using FlowJo v10 software.

Human embryonic genital ridges were collected from a week 7.0 male embryo in a dissection medium containing DMEM (Gibco), 10% FCS, and 1 mM sodium pyruvate (Sigma-Aldrich). The embryonic tissues were then dissociated by treating them with Collagenase IV (2.6 mg/ml) (Sigma-Aldrich) and 10 U/ml of DNase I (Sigma-Aldrich) in DMEM-F/12 (Thermo Fisher Scientific) and incubating them at 37°C for 10 min while being mixed by pipetting. The resulting cells were washed with FACS buffer, consisting of PBS (Thermo Fisher Scientific) with 3% FCS (Sigma-Aldrich) and 5 mM EDTA (Thermo Fisher Scientific), and then resuspended in 75 μl of FACS buffer. The cells were then stained with 0.5 μl of Alexa Fluor 488-conjugated anti-alkaline phosphatase (561495; BD Pharmingen) and 25 μl of PerCP-Cy5.5-conjugated anti-CD117 (333950; BD Pharmingen) for 15 min at room temperature. The samples were then washed with PBS (Thermo Fisher Scientific) and sorted using a SONY SH800 cytometer. The flow cytometry data were analysed using FlowJo v10 software.

## Immunofluorescence

EBs were first fixed with 4% PFA for 2 h at 4°C to preserve the protein structure. They were then embedded in O.C.T. compound (Cellpath) for frozen sections. The samples were incubated with primary antibodies specific for certain proteins, such as GFP, PRDM1, SOX17, TFAP2C, and OCT4, for either 1–2 h at room temperature or overnight at 4°C. Afterwards, the samples were incubated with fluorescent-conjugated secondary antibodies (Thermo Fisher Scientific) and DAPI for 1 h at room temperature. The primary antibodies used were anti-GFP (ab13970; Abcam), anti-PRDM1 (9115; Cell Signaling Technology), anti-SOX17 (AF1924; R&D Systems), anti-TFAP2C (sc-8977; Santa Cruz Biotechnology), anti-OCT4 (611203; BD Biosciences), anti-TFAP2A (sc-12726; Santa Cruz Biotechnology), anti-HAND1 (AF3168; R&D Systems), anti-GATA3 (ab199428; Abcam), and anti-NANOG (AF1997; R&D). Secondary antibodies used were donkey anti-goat IgG (H L), Alexa Fluor 647 (A21477; Thermo Fisher Scientific), donkey anti-rabbit IgG (H L), Alexa Fluor 568 (A10042; Thermo Fisher Scientific), donkey anti-mouse IgG (H+L), and Alexa Fluor 488 (A-21202; Thermo Fisher Scientific).

The samples were then imaged under a Leica SP8 upright or inverted scanning confocal microscope, which allows for high-resolution, three-dimensional imaging of the samples. In addition, the cells were cultured on an ibidi μ-slide and fixed in 4% PFA for 30 min at 4°C.

## 10X genomics

For each stage, a total of 5,000 cells were sorted using a SONY SH800 cytometer and were carefully gated to exclude any dead cells, debris or doublets. The sorted cells were then collected in an Eppendorf tube containing PBS (Thermo Fisher Scientific) with 0.04% weight/volume BSA (Thermo Fisher Scientific, 400 μg/ml). The sorted cells were then immediately processed using the 10x-Genomics Chromium single-cell 3' reagents kit v2 and pooled for

sequencing so that all lines would include all samples. The libraries were prepared according to the manufacturer's instructions and were sequenced on an Illumina HiSeq 4000, with a minimum coverage of 50,000 raw reads per cell. The sequencing was performed in a paired-end format (read 1: 26 cycles; i7 index: 8 cycles, i5 index: 0 cycles; read 2: 98 cycles).

## Bioinformatics

### 10X RNA sequencing processing

Multiplexed single-cell libraries were processed using the 10X Genomics Cell Ranger pipeline (v 3.0.2). Single-cell libraries were first processed using the 10x Genomics Cell Ranger pipeline. The cells were then subjected to quality control (QC) analysis to remove empty droplets and dead cells. After processing, libraries had the following QC metrics: 1.2–2.6 K cells per sample (aiming for 1.5 K recovery); median genes per cell 4,600–5,800; 90–95.6% mapped to the genome. Reads were aligned to a reference genome (*Homo sapiens* GRCh38) using STAR 2.5.4b (Dobin et al, 2013), and quantification of genes against an annotation reference (based on Ensembl GRCh38 v90).

### RNA sequencing processing

For SS2 reference datasets, cells were mapped as outlined in Bergmann et al (2022). Briefly, adapter reads were Trimmed using TrimGalore! and aligned to an appropriate reference genome (GRCh38/macFas5/Cjacchus3) using STAR aligner. Read counts were quantified using featureCounts 2.0.1. For 10X human datasets, reads were mapped with Cell Ranger pipelines (v 3.0.2) to GRCh38.

### Analysis

The initial analysis of the data was performed using Seurat version 3.1.4 (Stuart et al, 2019). The count data were normalised and scaled using NormalizeData based on log counts per 10,000 (logCP10k) and scaled using ScaleData. Clusters were then identified using FindCluster with a resolution of 0.1. Nearest neighbour graphs and UMAP plots were generated using the first 20 principal components. In addition, heatmaps of gene expression were created using the pheatmap package version 1.0.12, with cross-correlations calculated based on Pearson's correlation using R's inbuilt cor function and visualised using pheatmap.

### Integrative analysis

Individual datasets were first curated to remove preimplantation and extraembryonic tissues. The datasets were then integrated using logCP10k and the FindIntegrationMarkers method, with a total of 5,000 features and a k.filter of 50. In addition, the data were integrated using CCA and the first 20 principal components. Clustering of the integration-corrected gene expression matrices was performed using the FindClusters function, with the complexity parameter varied from 0.1–0.9 in increments of 0.1. For visualisation purposes, a complexity parameter of 0.9 was utilised in the figures presented in the study.

To initially establish the cell fate of the samples, key marker gene expression was plotted as a heatmap using the pheatmap tool. To confirm the validity of cell types across datasets, a scatter plot of differential expression was created. The x-axis displayed the logFC of a specific cluster in comparison with a reference cell type/cluster

(e.g., cluster 0 versus PSCs), whereas the y-axis displayed the same comparison in a second dataset. Genes in the top right and bottom left quadrants were considered conserved changes between the two datasets, whereas genes in the top left or bottom right quadrants represented changes specific to one dataset.

As a preliminary examination of individual bifurcations, diffusion maps were generated for selected sets of subclusters using the Destiny software (version 2.12.0) (Angerer et al, 2016) based on the integration-corrected expression matrices.

### Differential expression analysis

Unless otherwise indicated, differential expression between two groups was done in Seurat using MAST (Finak et al, 2015) (version 1.8.2). For volcano plots, genes were filtered to show genes with adjusted *P*-values <0.05 with a >1.2 FC.

### Network hubs and MA plots

Hub genes for given cell types were calculated based on log expression in specific clusters using arboreto v0.1.6 using GRNBoost2 followed by Google page rank-based prioritisation.

LogFC, adjusted *P*-values, and average expression values for the volcano and MA plots were calculated using Seurat and MAST and visualised using ggplot2. To simplify the number of genes shown, in Fig S2 We highlight genes based on the intersection between the top 100 hubs and DE genes. For other MA plots, we highlight a manually curated list of DE TFs.

### CellPhoneDB

Ligand–receptor interaction enrichment was calculated for individual samples using CellPhoneDB v4 (Garcia-Alonso et al, 2022) using the statistical_analysis method and visualised as dot plots for specific cell types.

### Mapping of cells from CS7 gastrula to embryoid bodies

Carnegie stage 7 human gastrula annotations were projected onto our EB dataset based on statistically enriched proximity in nearest neighbour graphs. Specifically, the aligned datasets were subsetted on the human CS7 gastrula and EB dataset and used to calculate a KNN graph (using the FindNeighbours function). For each cell within our EB dataset, the enrichment of individual CS7 gastrula annotations was calculated using a hypergeometric test, and final annotations were assigned based on adjusted *P*-values. In cases where multiple cell fates were enriched, cells were annotated with the lineage with the lowest adjusted *P*-value. Cells that showed no significant overlap in KNN graphs were not assigned a lineage.

### Mapping of cells to the CS6 marmoset embryo

Cells within our EB were mapped to the marmoset embryo based on proximity in KNN graphs in the CCA-aligned datasets. Aligned datasets were first subsetted on the marmoset dataset and EB dataset. For a cell, $j$, in the EB dataset, we calculated the KNN from the CS6 embryo, with positions at positions $\{r_1, r_2, .., r_K : r_i \in \mathbb{R}^3\}$, and calculated the shared nearest neighbour vector $\theta^{(j)} = \{\theta_1, \theta_2, .., \theta_K\}$. Weights were normalised $\hat{\theta}^{(j)} = \theta^{(j)}/c$, $c = \sum_i \theta_i$ and a projection of cell $j$ calculated as: $R = \sum_i r_i \hat{\theta}_i$, where $r_j \in \mathbb{R}^3$ denotes a three-dimensional position vector of marmoset cell $j$. After the mapping of individual cells, the density of specific groups, for example, PGCLCs (clusters 5 and 6), MeLC (clusters 7, 9 and 10), and basal (cluster 1), was calculated using the MATLAB function mvksdensity.

### Doublet detection

To minimise contamination from doublets in our analysis, we employed a strategy of limiting the number of cells loaded per chip, resulting in each sample capturing ~1,000–2000 cells. To identify potential doublets, we employed the DoubletFinder R package (McGinnis et al, 2019) for each sample. Given that each sample captured around 1,000 cells, we assumed a doublet rate of 1%, and for samples with ~2,000 cells, we assumed a doublet rate of 2%. We found that none of the clusters analysed in this study contained an excessive number of doublets.

### Waddington optimal transport analysis

Highly variable genes were computed across all single PSCs, PreME, and EB cells and used as input to PCA, with the first 50 PCs computed using irlba. The cells were assigned to clusters as described above, which were used as the basis for WOT. Transport maps were computed with parameters ($\lambda 1 = 1$, $\lambda 2 = 50$, $\varepsilon = 0.01$) between all pairs of time points using the PSCs as 0 h, PreME as 12 h, and all subsequent time points as $12 + t_i$ for $i \in \{1, 2, …, T\}$ and $T = \{12, 18, 24, 32, 40, 48, 96\}$. Ancestor contributions to populations at subsequent time points were estimated from these transport maps using the OT trajectory command-line interface function. Cell mass contributions between clusters across time points were concatenated into a cluster: time point X cluster: time point matrix, where the rows denote the contribution of cluster $j$ time point $i$ to cluster $j$ time point $i+1$. A power threshold ($P = 30$) was used to enforce sparsity on this matrix with values ≤ 0.1 censored to 0. This sparse matrix was then used as a weighted adjacency matrix to compute a directed KNN graph (k = 5), as shown in Fig S6. Meta-clusters were defined on this graph using the Walktrap community detection algorithm implemented in igraph, which were annotated based on the mean expression level of single cells that contribute to each original cluster (Fig S6). These annotations were then mapped back onto the original constituent single cells based on their cluster identity.

### Availability of materials

Any enquiries on reagents and cell lines can be directed to a.surani@gurdon.cam.ac.uk. Plasmids generated in this study will be made freely available upon request. Modified human embryonic stem cell lines generated in this study will be made available on request upon completion of a Materials Transfer Agreement.

# Data Availability

Single-cell RNA-seq (10X) data have been deposited at ArrayExpress under accession numbers E-MTAB-11283 and E-MTAB-11305. Code for repeating analyses will be available via a GitHub repository https://github.com/cap76/PGCLC.

# Supplementary Information

# Acknowledgements

This work was supported by the Wellcome Investigator Awards in Science (2094)75/Z/17/Z (to MA Surani), the Wellcome Investigator Awards in Science 096738/Z/11/Z (to MA Surani), the BBSRC research grant G103986 (to MA Surani), the Croucher Postdoctoral Research Fellowship (to WWC Tang), the Wellcome 4-Yr PhD Programme in Stem Cell Biology & Medicine (2038)31/Z/16/Z (to A Castillo-Venzor) and the Cambridge Commonwealth European and International Trust (to A Castillo-Venzor), the Isaac Newton Trust (to WWC Tang), the Butterfield Awards of Great Britain Sasakawa Foundation (to T Kobayashi and MA Surani), and the Astellas Foundation for Research on Metabolic Disorders (to T Kobayashi). The marmoset embryo research is generously supported by the Wellcome Trust (WT RG89228, WT RG9242), the Centre for Trophoblast Research, the Isaac Newton Trust, and JSPS KAKENHI 15H02360, 19H05759. TE Boroviak was supported by a Wellcome Sir Henry Dale Fellowship. JC Marioni acknowledges core support from EMBL and from Cancer Research UK (C9545/A29580), which supports MD Morgan. We would like to thank Roger Barker and Xiaoling He for providing human embryonic tissues and Charles Bradshaw for bioinformatics support. We also thank The Weizmann Institute of Science for the WIS2 human PSC line and the Genomics Core Facility of CRUK Cambridge Institute for sequencing services. We thank members of the Surani laboratory for insightful comments and critical reading of the manuscript.

## Author Contributions

A Castillo-Venzor: conceptualization, data curation, formal analysis, funding acquisition, validation, investigation, visualization, project administration, and writing—original draft, review, and editing.
CA Penfold: conceptualization, resources, data curation, software, formal analysis, supervision, validation, investigation, visualization, methodology, project administration, and writing—original draft, review, and editing.
MD Morgan: formal analysis, investigation, methodology, and writing—review and editing.
WWC Tang: conceptualization, investigation, and writing—review and editing.
T Kobayashi: resources, investigation, and writing—review and editing.
FCK Wong: investigation and writing—review and editing.
S Bergmann: resources, investigation, and writing—review and editing.
E Slatery: resources, investigation, and writing—review and editing.
TE Boroviak: resources, supervision, investigation, and writing—review and editing.
JC Marioni: resources, supervision, investigation, and writing—review and editing.
MA Surani: conceptualization, resources, supervision, funding acquisition, investigation, project administration, and writing—original draft, review, and editing.

## Conflict of Interest Statement

WWC Tang is currently employed by Adrestia Therapeutics Ltd. The other authors declare that they have no conflict of interest.

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
