## [Reviewer comments · Life Science Alliance]

Life Science Alliance

Origin and segregation of the human germline

Aracely Castillo-Venzor, Christopher Penfold, Michael Morgan, Walfred Tang, Toshihiro Kobayashi, Frederick Wong, Sophie Bergmann, Erin Slatery, Thorsten Boroviak, John Marioni, and M. Azim Surani

DOI: <https://doi.org/10.26508/lsa.202201706>

Corresponding author(s): Dr. Azim Surani (Wellcome/Cancer Research UK Gurdon Institute) and Aracely Castillo-Venzor

Review Timeline:

Submission Date:	2022-09-03
Editorial Decision:	2022-09-05
Revision Received:	2023-02-11
Editorial Decision:	2023-03-08
Revision Received:	2023-04-10
Editorial Decision:	2023-04-11
Revision Received	2023-05-10
Accepted:	2023-05-11

Scientific Editor: Eric Sawey

Transaction Report:

Please note that the manuscript was previously reviewed at another journal and the reports were taken into account in inviting a revision for publication at *Life Science Alliance* prior to submission to *Life Science Alliance*.

No Peer Review Process File is available with this article, as the authors have chosen not to make the review process public in this case.

Re: Life Science Alliance manuscript #LSA-2022-01706-T

Dr. Azim Azim Surani
Wellcome Trust/Cancer Research UK Gurdon Institute
Wellcome Trust/Cancer Research UK Gurdon Institute
Tennis Court Road
Cambridge, Cambridgeshire CB2 1QR
United Kingdom

Dear Dr. Surani,

Thank you for submitting your manuscript entitled "Origin and segregation of the human germline" to Life Science Alliance. The manuscript was assessed by expert reviewers at EMBO Reports and then transferred to LSA. We are interested in these findings, and would like to invite further consideration of this manuscript at LSA pending the following revisions:

- Address Reviewer 1's comments via added Discussion or clarification. Requests for additional experimentation are not required.
- You may respond and add Discussion or clarification points in the text in response to Reviewer 2, if you wish.
- Address Reviewer 3's comments, excluding point #9.

Thank you for this interesting contribution to Life Science Alliance. We are looking forward to receiving your revised manuscript.

Sincerely,

- A letter addressing the reviewers' comments point by point.
- An editable version of the final text (.DOC or .DOCX) is needed for copyediting (no PDFs).
- High-resolution figure, supplementary figure and video files uploaded as individual files: See our detailed guidelines for preparing your production-ready images, <https://www.life-science-alliance.org/authors>

B. MANUSCRIPT ORGANIZATION AND FORMATTING:

Re: Life Science Alliance manuscript #LSA-2022-01706-TR

Dr. Azim Surani
Wellcome/Cancer Research UK Gurdon Institute
Wellcome Trust/Cancer Research UK Gurdon Institute
Tennis Court Road
Cambridge, Cambridgeshire CB2 1QR
United Kingdom

Dear Dr. Surani,

Thank you for submitting your revised manuscript entitled "Origin and segregation of the human germline" to Life Science Alliance. The manuscript has been seen by the original reviewers whose comments are appended below. While the reviewers continue to be overall positive about the work in terms of its suitability for Life Science Alliance, some important issues remain. We therefore encourage to address the final Reviewer 1's points.

Our general policy is that papers are considered through only one revision cycle; however, given that the suggested changes are relatively minor, we are open to one additional short round of revision. Please note that I will expect to make a final decision without additional reviewer input upon resubmission.

Please submit the final revision within one month, along with a letter that includes a point by point response to the remaining reviewer comments.

- A letter addressing the reviewers' comments point by point.
- An editable version of the final text (.DOC or .DOCX) is needed for copyediting (no PDFs).
- High-resolution figure, supplementary figure and video files uploaded as individual files: See our detailed guidelines for preparing your production-ready images, <https://www.life-science-alliance.org/authors>
- Summary blurb (enter in submission system): A short text summarizing in a single sentence the study (max. 200 characters including spaces). This text is used in conjunction with the titles of papers, hence should be informative and complementary to the title and running title. It should describe the context and significance of the findings for a general readership; it should be written in the present tense and refer to the work in the third person. Author names should not be mentioned.

B. MANUSCRIPT ORGANIZATION AND FORMATTING:

Sincerely,

RE: Life Science Alliance Manuscript #LSA-2022-01706-TRR

Dr. Azim Surani
Wellcome/Cancer Research UK Gurdon Institute
Wellcome Trust/Cancer Research UK Gurdon Institute
Tennis Court Road
Cambridge, Cambridgeshire CB2 1QR
United Kingdom

Dear Dr. Surani,

Thank you for submitting your revised manuscript entitled "Origin and segregation of the human germline". We would be happy to publish your paper in Life Science Alliance pending final revisions necessary to meet our formatting guidelines.

- please upload your manuscript text as an editable doc file
- please add ORCID ID for first corresponding author-you should have received instructions on how to do so
- please consult our manuscript preparation guidelines <https://www.life-science-alliance.org/manuscript-prep> and make sure your manuscript sections are in the correct order
- please add a summary blurb/alternate abstract to our system
- please add the Twitter handle of your host institute/organization as well as your own or/and one of the authors in our system
- please add a conflict of interest statement to the main manuscript text
- please use the [10 author names, et al.] format in your references (i.e. limit the author names to the first 10)
- please add a callout for Figure S2A-B and S2E,F,I; Figure S6A-D and S6H-I to your main manuscript text

Figure Check:

- please refer to the panels for Supplementary Figure 7 A-I, Supplementary Figure 5 and Figure S3C legends
- please add scale bars to Figure 4F

To upload the final version of your manuscript, please log in to your account: <https://lsa.msubmit.net/cgi-bin/main.plex>

A. FINAL FILES:

B. MANUSCRIPT ORGANIZATION AND FORMATTING:

Sincerely,

RE: Life Science Alliance Manuscript #LSA-2022-01706-TRRR

Dr. Azim Surani
Wellcome/Cancer Research UK Gurdon Institute
Wellcome Trust/Cancer Research UK Gurdon Institute
Tennis Court Road
Cambridge, Cambridgeshire CB2 1QR
United Kingdom

Dear Dr. Surani,

Thank you for submitting your Research Article entitled "Origin and segregation of the human germline". It is a pleasure to let you know that your manuscript is now accepted for publication in Life Science Alliance. Congratulations on this interesting work.

DISTRIBUTION OF MATERIALS:

Again, congratulations on a very nice paper. I hope you found the review process to be constructive and are pleased with how the manuscript was handled editorially. We look forward to future exciting submissions from your lab.

Sincerely,
